

# An Improved Method Based on VGGNet for Refined Bathymetry from Satellite Altimetry: Reducing the Errors Effectively

**Xiaolun Chen[1], Xiaowen Luo[1,4], Ziyin Wu[1,2,3], Xiaoming Qin[3], Jihong Shang[1], Mingwei Wang[1], Hongyang Wan[1]**

[1]Key Laboratory of Submarine Geosciences, Second Institute of Oceanography, Ministry of Natural Resources, Hangzhou, China
[2]School of Oceanography, Shanghai Jiao Tong University, Shanghai, China
[3]Ocean College, Zhejiang University, Zhoushan, China
[4]Key Laboratory of Ocean Space Resource Management Technology, Marine Academy of Zhejiang Province, Hangzhou, China

**Correspondence**: Xiaowen Luo (cdslxw@163.com); Ziyin Wu (zywu@vip.163.com)

**Abstract.** At present, only approximately 10% of the global seafloor topography has been finely modeled, and the rest are either lacking in data or not accurate enough to meet practical requirements. On the one hand, satellite altimeter has the advantages of large-scale and real-time observation, thus is widely used in the measurement of bathymetry, the core of seafloor topography. However, there is often room for improvement in its precision. On the other hand, multibeam echosounder bathymetric data is highly precise but normally limited to a smaller coverage, which forms a complementary relationship with the bathymetry derived from satellite altimetry. To combine the advantages of satellite altimetry-derived and multibeam sonar-derived bathymetry, we apply deep learning, which is powerful in the field of digital image automation, to perform multibeam sonar-based bathymetry correction for satellite altimetry bathymetry data. Specifically, we modify and improve a pretrained VGGNet neural network model with a depth of 19 layers to train on three sets of bathymetry data from the West Pacific, Southern Ocean, and East Pacific, respectively. Experiments show that the correlation of bathymetry data before and after correction can reach a high level, with the performance of $R^2$ being as high as 0.81 and the RMSE improved over 19% compared with previous research. We then explore the relationship between $R^2$ and water depth and conclude that it varies at different depths and thus the terrain specificity was a factor that affects the precision of correction. Finally, we use the difference of water depth before and after the correction to evaluate the correction results, and find that our method can improve by more than 17% compared with previous research. The results show that using the deep learning VGGNet model can better perform the correction of the bathymetry derived from satellite altimetry, thus providing a method for accurate modeling of the seafloor topography.



## 1 Introduction

Submarine topographic survey is a basic marine surveying and mapping work, whose purpose is to obtain the three-dimensional coordinates of submarine topographic points, including measurement position, water depth, water level, sound speed, attitude, azimuth and other information, the core of which is water depth measurement. Modern multibeam sounding systems began to rise in the 1960s. Fox et al. (1992) conducted a quantitative analysis of the changes in the submarine topography caused by the submarine volcanic eruption based on the multibeam sonar data and the submarine robot's measured images. Wu (2001) put forward the key statistical parameters to attain the seafloor tracking of the multibeam sounding system and established the mathematical model and expert system for real-time tracking of the seafloor terrain. Schimel et al. (2015) analyzed the continuous observation of multibeam data and found that the uncertainty information provided by the multibeam processing algorithm CUBE can be used to better calculate the displacement of the sediment volume. Ma et al. (2006) found that full coverage and high-efficiency multibeam sonar can be combined with side-scan sonar, which has good complementarity when detecting submarine targets, and can improve the accuracy of target recognition. Ji (2017) applied backpropagation (BP) neural network to build a feature database of seabed terrain based on multibeam data to attain automatic classification of seabed terrain complexity. Pike et al. (2019) combined Pleiades multispectral imagery and multibeam data to measure the water depth of two shallow waters in the northeastern Caribbean. Cooper et al. (2021) proposed a method that uses small unmanned aerial vehicle (sUAV) photogrammetry as well as multibeam sonar data to generate a complete bathymetry map of a reservoir.

The multibeam sounding method has the advantage of high spatial precision, which enables the underwater sounding mode to achieve a high-quality leap from point to line and from line to surface (Li, 1999). However, with the low efficiency, high cost and long measurement time required, these shortcomings make it difficult to conduct submarine surveys in a wide range of sea areas. Thus, the coverage of shipborne soundings is still very sparse at present. It's estimated that only about 10% of the global sea area is covered with shipborne survey data, and a considerable part of it, especially in the deep ocean areas, consists of analog signals from 1950 to 1967, whose accuracy is relatively low (Becker et al., 2009).

Satellite altimetry is a space measurement technology that uses artificial satellites as a carrier to measure the distance of the satellite from the surface of the earth using radar, laser, and other ranging technologies, to obtain the surface terrain of the earth, through which a gravity field model and terrain features of the ocean can be constructed. Parker (1972) derived the expression of gravity in the frequency domain and put forward the material interface of the model of abnormal gravity changes caused by fluctuations, which laid the foundation for the development of seafloor topography inversion. Since the launch of the Seasat in 1978, many researchers have used satellite altimetry data to model water depth, such as Dixon et al. (1983), Smith and Sandwell (1994), Ramillien and Cazenave (1997), and Arabelos (1997). Calmant and Baudry (1996) provided a comprehensive overview of the techniques and data used in





bathymetric models. Yeu et al. (2018) combined multibeam sonar, satellite altimetry-derived gravity anomalies and airborne LiDAR data and managed to effectively improve the accuracy of water depth measurement for up to 0.2 m in shallow waters less than 5 m. Brêda et al. (2019) introduced and evaluated several data assimilation (DA) methods for satellite altimetry data, which has reduced the biased bathymetry errors in the hydrodynamic model for up to 65% compared to past observations while at the same time increased the optimizer runtime to 103 times. Wölfl et al. (2019) summarized the significance, technology, data sources, development, and challenges of global seafloor topography surveys and researches and proposed recommendations for the goal of a precise global bathymetry map inspired by GEBCO Seabed 2030 Project. Sepúlveda et al. (2020) established a sea depth uncertainty model for satellite altimetry, quantified the high-wavenumber content within the satellite-derived data and proved the model in the bathymetry generated from the forecast of tsunami, with certain parameters varied regionally.

The emergence of satellite altimetry has made seabed topography measurement no longer limited to the shipborne sonar, and has provided new technical means for large-scale, real-time global measurement. However, existing researches have shown that compared with the multibeam-derived bathymetry, it still has the limitation on spatial resolution and thus is influenced by submarine parameters such as depth, surrounding topography, computational scales and so on (Dierssen et al., 2020; Dettmering et al., 2020; Wu et al., 2021).

In recent years, deep learning has become an important scientific computing tool and made great contributions and development in various aspects such as image classification (Mou et al., 2017; Li et al., 2019; Hong et al., 2021), object detection (Girshick et al., 2014), feature extraction (Evans and Ruf et al., 2021), etc., making multisource big data-based ocean observations available and efficient and consequently being applied to the field of seafloor topography inversion. Jena et al. (2012) developed an artificial neural network (ANN) model based on radial basis function (RBF) to predict the water depth based on satellite-derived gravity data, with the results demonstrating that the precision of the ANN model is higher than other submarine topography models. Jha et al. (2013) used the geostatistical direct sampling (DS) based multi-point statistics (MPS) algorithm, merging the low-frequency high-resolution multibeam sonar data and high-frequency low-coverage shipborne survey data, utilizing the former to provide prior constraining information to simulate and generate fine depth maps. Moran (2020) discussed the global viability of machine learning models for inversing bathymetry and the probability of an enhanced global model by experiment and concluded machine learning could help with the determination of a decision boundary when generating models. Ghorbanidehno et al. (2021) introduced a principal component analysis (PCA) connected deep neural network (DNN) to perform bathymetry inversion using flow velocity observations, proving its accuracy and availability for a high-dimensional riverbed topography model with sparse measurements.

By attaining the unification of the spatial resolution of multibeam data and the spatial coverage of satellite altimetry data, it can provide a new means for high-



precision, real-time global seafloor topography surveying. In this paper, we proposed a
novel optimization algorithm based on VGGNet, a model for application of
convolutional neural network (CNN), aiming to enhance the precision of satellite
altimetry-derived bathymetry, which mostly lies on the range of the estimated average
of global ocean depth, by the input of multibeam sonar bathymetry data (Charette et al.,
2010).

The main contributions of this article are as follows.
1. A data combination of high-spatial-resolution multibeam sonar-derived
bathymetry (truth data) and high-coverage satellite altimetry-derived bathymetry (to-
be-corrected data) is synthesized to obtain a corrected version of the latter, with the
advantage from both sides.
2. A convolutional neural network (CNN) based VGGNet algorithm framework is
for the first time proposed to compute the distance (loss) between the two inputs - to-
be-corrected data and truth data, where the former is transformed by minimizing the
distance between them with backpropagation, generating an image that best match the
latter.
3. Experiments are conducted in West Pacific, Southern Ocean, and East Pacific,
to test the performance of the algorithm, with the results showing that the improvement
in computational precision can be over 17% in comparison with previous researches as
far as we conclude.
The rest of the article is organized as follows. An introduction to the background
of VGG-19 framework and the methodology of the correction of satellite altimetry-
derived bathymetry data using multibeam sonar data is elaborated in Section II. The
neural network experiments and their results are presented in Section III. Finally,
Section IV concludes the article.

## 2 Proposed Methodology

In this section, we elaborate the related background of the CNN-based VGGNet
(VGG-19) algorithm and its detailed application to the correction of the bathymetry
data. As shown in Figure 1, the structure of the proposed network consists of mainly
three parts: 1) the input of the truth and to-be-corrected bathymetry data; 2) the
designation of network model, requiring a pretrained VGG-19 framework, a loss
function, gradient descent, and the optimization loop; 3) the output of the corrected
version of satellite altimetry-derived bathymetry data.

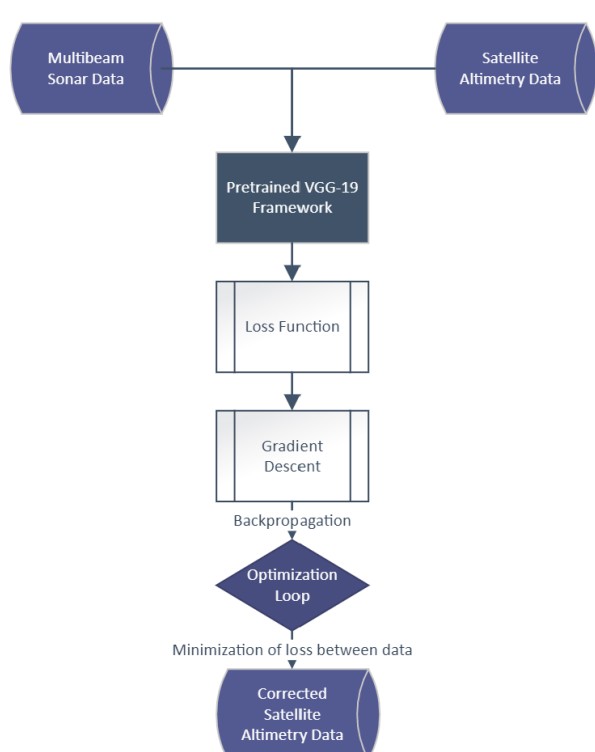

Fig.1 Main structure of the proposed network.

**2.1 Framework of VGG-19**

The convolutional neural network (CNN) models have been improved and updated
for better application of large-scale image recognition, such as AlexNet (Krizhevsky et
al., 2012), CaffeNet (Jia et al., 2014), LeNet (LeCun et al., 1998) and VGGNet
(Simonyan et al., 2014), etc. Compared with most previous CNN-originated models
that have 4 – 7 layers, VGG-19, a form of VGGNet, is constructed by 19 layers, which
includes 16 convolutional layers and 3 fully connected layers, enabling it to extract the
more abstract and deeper image features and reduce the amount of parameters while
still retain the same receptive field, thus has improved the efficiency and accuracy of
image computing (Huo et al., 2020; Islam et al., 2020; Schulz et al., 2020).
The structure of VGG-19 is displayed in Figure 2. The entire network uses the same
size of convolution kernels (3x3) and maximum pooling kernels (2x2). The
combination of several small filter (3x3) convolutional layers is better than a large one
(5x5 or 7x7) in the previous models. Since the convolution kernel focuses on expanding
the number of channels and the pooling kernel focuses on reducing the width and height,
the architecture is deeper and wider while the increase of calculation slows down,
showing the network a larger receptive field. At the same time, the network parameters
are reduced, and the ReLU (Rectified Linear Unit) activation function is used multiple
times to create more linear transformations to enhance the learning ability[40].




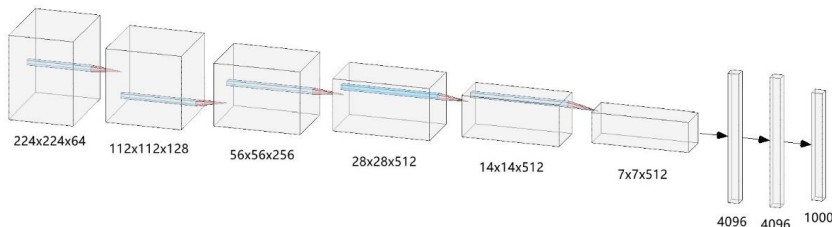

224x224x64  112x112x128  56x56x256  28x28x512  14x14x512  7x7x512  4096  4096  1000

Fig. 2 Architecture of VGGNet model used in this paper. The boxes represent the size of each
layer.

**2.2 Model Training Steps**

The correction of the bathymetry is conducted under the model of VGG-19. The principle of the correction model is to define a distance function that describes how different the two input images are. The multibeam-derived data image and the satellite altimetry-derived data image covering the same area are passed to the model, which is supposed to return the intermediate layer outputs from the model. The distance function $L$ that we use is shown below:

$$L^l(x,p) = \sum_{i,j} \left( F_{ij}^l(x) - P_{ij}^l(p) \right)^2 \tag{1}$$

where $x$ stands for the multibeam sonar-derived bathymetry image, $p$ stands for the satellite altimetry-derived bathymetry image, and $i,j$ stand for the serial number of pixel points of the input images. Let $V_{nn}$ be a pre-trained VGG-19 network and $X$ be any image, then $V_{nn}(X)$ is the network fed by $X$. Let $F_{ij}^l(x) \in V_{nn}(x)$ and $P_{ij}^l(p) \in V_{nn}(p)$ describe the respective intermediate feature representation of the network with the inputs $x$ and $p$ at layer $l$. At last, optimizers update rules are applied to iteratively update the images, which minimize a given loss with respect to the inputs.

The evaluation of the precision of correction is based on the comparisons with previous study. In order to quantify the differences and connections between the predicted value and truth value, here we choose two evaluation measurement, root mean square error (RMSE), normalized RMSE (NRMSE), and coefficient of determination ($R^2$), as follows respectively:

$$RMSE = \sqrt{\frac{\sum_{i=1}^{n}(\widehat{f_i} - y_i)^2}{n}} \tag{2}$$

$$NRMSE = \frac{RMSE}{y_{max} - y_{min}} \tag{3}$$



$$R^2 = 1 - \frac{\sum_{i=1}^{n}(y_i - f_i)^2}{\sum_{i=1}^{n}(y_i - \overline{y})^2} \tag{4}$$

where $n$ represents the number of the values from dataset, $i$ represents the serial number of the value from the dataset, $f$ represents the predicted values and $y$ represents the truth values. The normalization of RMSE can make data sets of different numerical ranges easier to compare. $NRMSE$ and $R^2$ normally range from $0 - 1$. The smaller $RMSE, NRMSE$ and the bigger $R^2$ mean the higher correlation between the datasets.

Using the multibeam-derived data as the content image to match, we input and transformed the satellite altimetry-derived data under the framework of VGG-19 to minimize the losses and distances between them so that we could attain an improved bathymetry data that combined the advantages of both – the high spatial precision of multibeam data and the wide spatial coverage of satellite data.

**3 Experiments and Results**

**3.1 Experiment Data**

The original shipborne multibeam sonar bathymetry data used in the experiment is acquired at NOAA National Geophysical Data Center (2009). The interpolation preprocessing on the raw data is carried out to output the gridded digital elevation model (DBM) data. Meanwhile, the satellite altimetry data used in the experiment is acquired and extracted from NGDC's ETOPO1 1 arc-minute global relief model, clipped with the same range as that of the multibeam sonar data above (NOAA National Geophysical Data Center, 2004). The grid resampling of the satellite altimetry data is performed according to the resolution of the corresponding multibeam data, in order to unify the resolutions of the pairs to facilitate subsequent operations.

We use a total of three pairs of multibeam-satellite bathymetry data respectively from the West Pacific, Southern Ocean, and East Pacific, and conduct experimental analysis. The location and parameters of the data are shown in Figure 3 and Table 1.

For the VGG-19 model, the input parameter is a pair of multibeam-satellite bathymetry data, and the output parameter is the corrected satellite altimetry data. In the dataset, 50% of them are randomly selected as the training set to initially fit the model and update the parameters, and the remaining 50% are created as the validation set to provide an unbiased evaluation of the model fitted on the training set, which is the prediction results.





Fig. 3 Location of the bathymetry data in (a) West Pacific, (b) Southern Ocean, and (c) East
Pacific.
Table 1 The parameters of bathymetry data.

| | Grid resolution (m) | Dataset size | Area (km²) | Depth range (m) |
|---|---|---|---|---|
| **West Pacific** | 103 | 12,624,868 | 133,937 | -8,987 – -369 |
| **Southern Ocean** | 93 | 5,097,104 | 43,700 | -4,077 – -211 |
| **East Pacific** | 93 | 9,135,007 | 78,318 | -3,921 – -1,266 |


**3.2 Analysis of Results**

The output of the deep learning model is the corrected satellite altimetry bathymetry
data. Under the processing of the VGG-19 model, the surface texture of the satellite
altimetry-derived seabed topography from West Pacific, Southern Ocean and East
Pacific has been refined, and the water depth range has been corrected, resulting in a
reduction in the distance from the truth value.

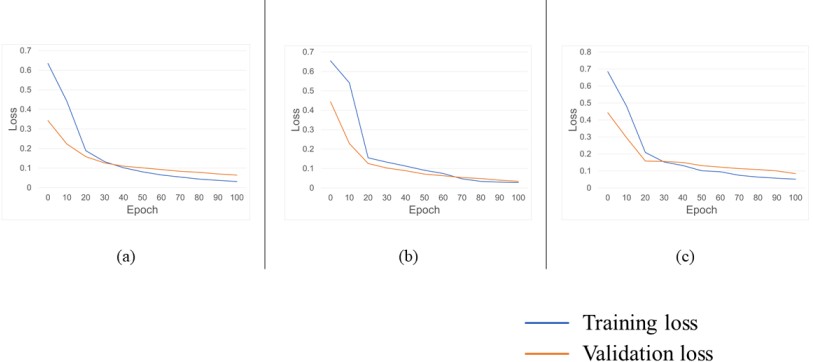

(a)  (b)  (c)

—— Training loss
—— Validation loss


Fig.4 The loss of the training set and validation set from the model of (a) West Pacific, (b)
Southern Ocean and (c) East Pacific.


The loss function is used to estimate the gap between the output value of the model
and the truth value to guide the subsequent optimization steps of the model. The smaller
the loss function value, the better the model. The loss on training and test sets are shown
in Figure 4. In the three experimental areas, the loss of the model has dropped sharply
to around 0.2 after 20 epochs, and starts to decrease gradually, especially after 70 epochs.
Moreover, it shows that no obvious overfitting phenomenon is found during the
computing process. It can be concluded that the machine learning of the VGG-19 model
can effectively reduce the loss for the experimental data from the three sea areas.
The parameters of the performance of the model are evaluated by running tests on
50% of the multibeam sonar data from validation set, with its outcome listed in Table
2. From the perspective of $R^2$, there is a high correlation between the corrected datasets
from the West Pacific, Southern Ocean and East Pacific, respectively 0.80, 0.81 and
0.77, and the truth datasets, indicating an excellent fit. In terms of RMSE and NRMSE,
the figures show that the correction algorithm results in errors of 267 meters, 102 meters,
and 87 meters in the Western Pacific, Southern Ocean, and Eastern Pacific datasets,
along with NRMSE being 0.031, 0.026, and 0.033, respectively. Compared with
previous similar studies [26] [29], our algorithm is able to improve the NRMSE of the
datasets by more than 19%, proving its potential. In addition, there is a consistent trend
in the changes of $R^2$ and RMSE, with the correction effect of the data in Southern Ocean
is the best, followed by the West Pacific, and then the East Pacific.

Table 2 The precision of satellite altimetry correction.

|  | $R^2$ | RMSE (m) | NRMSE |
|---|---|---|---|
| **West Pacific** | 0.80 | 267 | 0.031 |
| **Southern Ocean** | 0.81 | 102 | 0.026 |





| | | | |
|---|---|---|---|
| **East Pacific** | 0.77 | 87 | 0.033 |


In experiments, we find that the precision of correction, taking $R^2$ as an example,
varies with water depth, as shown in Figure 5. As can be seen from the figure, in general,
the minimum of $R^2$ is above 0.2, which occurs at the extreme value of water depth,
while the maximum can reach more than 0.9 and the water depth in each water area
varies, with maximum and minimum values for each sea area being almost identical. In
the West Pacific data, $R^2$ is higher than 0.8 in the water depth range of about -4,500 to
-1,900 meters, showing a strong correlation, with a maximum at about -3,200 meters.
For the Southern Ocean data, $R^2$ is strongly correlated at around -500 m and around -
1,800 m to -2,400 m, with a maximum around -2,200 m. For the eastern Pacific data,
$R^2$ is strongly correlated in the range of about -2,400m to -3,600m, with a maximum
around -3,500m.
According to experiences, the precision of machine learning is positively correlated
with the volume of data in the samples from dataset. Without considering other
parameters, the larger the sample size, the higher the learning precision tends to be, and
vice versa. In this experiment, this theory has also been verified. Combined with the
histogram of water depth values, the depths where the distribution of bathymetry data
points are scattered and the variance is large are often the areas where $R^2$ shows a low
level, and the depths with high $R^2$ often also have more concentrated distribution and
small variance. Specifically, at the maximum and minimum values of the water depths
in these three sea areas, due to the small amount of sample data, the precision of
machine learning is also low. Precision in the depth range where the largest sample size
is distributed is highly correlated. The rise and fall of the $R^2$ value curves in the figure
at certain water depths also reflect the particularity of the distribution of the bathymetry
values of the local seabed topography to a certain extent. Experiments show that with
the input of sufficient data volume, the satellite altimetry-derived bathymetry data
corrected by the VGG-19 model can be highly fitted with the multibeam-derived data
in specific water depth ranges.

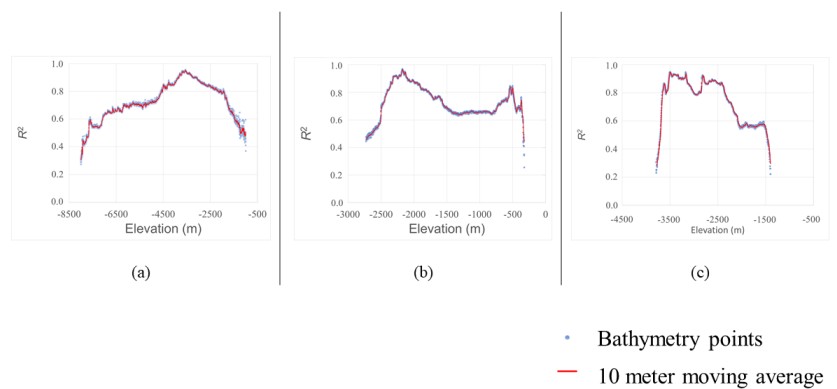




Fig.5 Relationship between water depth and precision ($R^2$) in (a) West Pacific, (b) Southern Ocean and (c) East Pacific.

We subtract the corrected water depth value of satellite altimetry data from the truth value of multibeam sonar data and find that the distribution of errors between the two is in the form of high in the middle (zero) and low on both sides, that is, the closer the error is to 0, the greater the number of data points, and vice versa, as shown in Figure 6. In the West Pacific data, the data point with zero error as the maximum value is isolated, not continuous with the rest of the curve, indicating that the algorithm results in significantly more error-free bathymetry points. In the other two data, the data curves are relatively continuous, decreasing from the maximum value of zero to both sides, while the curve of the Southern Ocean data is more convergent near zero than the East Pacific one, indicating that its correction effect is better.

For a more intuitive representation, we use the absolute value of the results above to calculate the percentage of the data within the range of 2%, 1% and 0.5% to the total depth of each data, with the values representing the errors from the truth value, as listed in Table 3. As the error range decreases, the number of data points increases gradually. On average, the data points with an error within the range of 2% of the depth value account for 70.58% of the total, 49.21% within the 1% range, and 30.01% within the 0.5% range. Compared with previous studies, the correction precision of the deep learning VGG-19 model can be effectively improved by over 17%.

Among the depth range indicators, the accuracy of the corrected Southern Ocean data is consistently better than the other two by a relatively large margin. In the 2% range, the East Pacific and West Pacific data performed almost indistinguishably, with the East Pacific data slightly higher. Under the strictest standard of 0.5% range, the performance results of the two are widened, with the West Pacific data being better. Combined with Table 2, it can be found that the changes of the parameters in the two tables show relative consistency, with the data of the Southern Ocean having the best correction effect, while the data of the West Pacific and East Pacific being the second and lowest respectively in most cases.





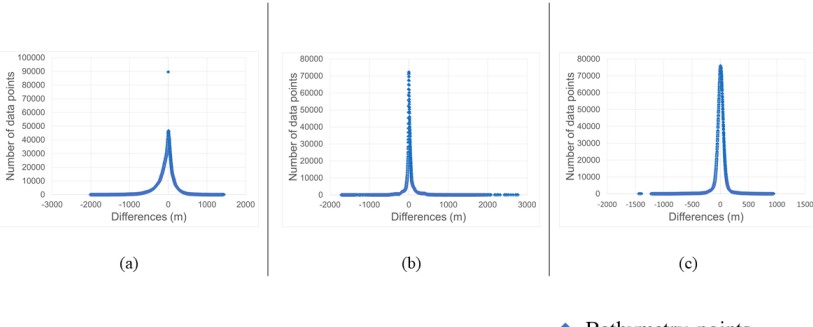

♦ Bathymetry points


Fig.6 Differences between corrected and truth values in (a) West Pacific, (b) Southern Ocean and
(c) East Pacific.

Table 3 Proportion of corrected errors from truth values within 2%, 1% and 0.5% depth range.

|  | 2% of depth (%) | 1% of depth (%) |
|---|---|---|
| **West Pacific** | 67.25 | 45.73 |
| **Southern Ocean** | 76.19 | 60.34 |
| **East Pacific** | 68.30 | 41.55 |


**4 Conclusions**
In this study, we propose a deep learning-based VGGNet pretrained algorithm
model to correct the satellite altimetry-derived bathymetry data with multibeam sonar-
derived bathymetry as truth data. The core idea of the correction model is to define and
minimize the distance (loss) between the truth data and the data to be corrected and
finally output the corrected satellite altimetry seafloor topography accordingly. We then
evaluate the model performance using three pairs of bathymetric data from the West
Pacific, Southern Ocean, and East Pacific. In the process of testing, the loss of training
set and validation set of the data has been effectively reduced, which proves the
effectiveness of the model.
We selected three indicators, $R^2$, RMSE and its derived NRMSE to evaluate the
correction results of the data, showing excellent outcomes and the NRMSE indicator
being over 19% higher than previous research. Further, by analyzing the difference of
$R^2$ values at different water depths, we find that the correction precision of deep learning
has a positive correlation trend with the sample size, that is, the accuracy of the depth
values with more data points is higher, and vice versa. Finally, after finding that the
difference between the truth value and the corrected value gradually decreases from the


maximum value at zero to both sides of the number axis, we analyze the proportion of the absolute value of the difference to the overall water depth and find that our model can improve the correction precision by more than 17% comparing with previous research. Overall, among the three test areas, the Southern Ocean data has the highest correction precision, followed by the West Pacific data, and the East Pacific data ranked last.

*Code Availability*. The raw code and part of the demonstration data of the model involved in this research have been archived at https://doi.org/10.5281/zenodo.6769649 (Chen et al., 2022).

*Author contributions*. XC and XL, ZW conceived the research. XC carried out the experiments and led the writing of the paper. XQ offered guidance to the modeling process. JS, MW and HW provided datasets for the experiment.

*Competing interests*. The contact author has declared that neither he nor their co-authors have any competing interests.

*Acknowledgements*. This research is funded by the following project: 1) National Natural Science Foundation of China (*41830540*). 2) National Key Research and Development Program of China (*2020YFC1521700* and *2020YFC1521705*). 3) The Open Fund of the East China Coastal Field Scientific Observation and Research Station of the Ministry of Natural Resources (*ORSECCZ2022104*). 4) The Deep Blue Project of Shanghai Jiao Tong University (*SL2020ZD204*). 5) Independent Project of SOED State Key Laboratory of the Second Institute of Oceanography, Ministry of Natural Resources (*SOEDZZ2101*). 6) Zhejiang Provincial Project (*330000210130313013006*). We express our most sincere gratitude to the above funds that enable our research.

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
