# Peer review of "An Improved Method Based on VGGNet for Refined Bathymetry"

_Geoscientific Model Development, 2022_

## Referee Comment (RC1)

Comments:

The authors refined the bathymetry from satellite altimetry based on an improved VGGNet. While this work is motivated, the manuscript suffers some technical problems.

Major comments:

As a start, it is unclear what the motivation for the choice of the machine learning approach used by VGGNet is. Why did the authors not choose another method for comparison in performance, given that they got poor results from this approach? The author's less innovative approach to VGGNet modification is not in line with the "An Improved Method" mentioned in the article.

The overall structure of the manuscript is unclear. The authors presented some details of the deep learning methods in both the introduction and methods. The multibeam-satellite data and satellite altimetry data are not well described, for example, the data coverage both spatially and temporally, and why they are used. Geographic data from different sources should be preprocessed to eliminate the effect of coordinate errors. What is the specific method of interpolation preprocessing in the article? Why was the method selected? The input and output data of the model are not reasonable. The satellite altimetry data should be used as an input parameter, with the true multibeam satellite bathymetry data being the expected result.The model's experimental testing component should be assigned to one of the three datasets, or to the non-training and validation part of the three regions.

The format of the reference is incorrect. In line 130, the authors cite (Charette et al., 2010), but I did not find any correlation between the authors' method and Charette's theory of the volume of the earth. In section 3.1, the original data source references and links are required. In line 278, the [26][29] method of marking the literature does not fulfill the criteria of uniform reference labeling. There were no similar findings between reference [26] and the approach described in this article. In line 333, what are the previous studies mentioned here? The literature and comparative data need to be annotated.

The manuscript is in need of editing by either a technical writer or, at least, someone with a technical background and conversant with native English. There are many instances of using English words that would better describe the situation than those used.

Specific comment:
L29 and L278: Is it RMSE metric improvement or NRMSE?
L182: What is the meaning of the upper mark 40?
Fig.4, Fig.5 and Fig.6: These figures are too tiny to meet GMD's figure content guidelines requirements.

---

## Author Comment (AC2)

Dear Referee:

We are glad to receive your comments on our manuscript. We revised the text in line with your comments and responded to some of your questions based on our humblest opinion, as described below accordingly:

1. First, the main reason we chose to use VGGNet in this study is because the basic idea of our research is very similar to *neural style transfer*, a common image style transfer processing algorithm in popular culture, which is built on the VGGNet framework. Second, in searching for relevant literature on seafloor topography inversion involving machine learning, we found several target literatures that can be compared with our method in accuracy (all listed in ref.). Through comparison, we found that our method using VGGNet can make the inversion accuracy higher than the RBF, MPS and other algorithm models used in previous studies. Finally, we have taken on board your comments and revised the title of the manuscript. Our original idea was to emphasize that this was an "improved" approach to accuracy.

2. In **Table 1**, we add the latitude and longitude information of the center points of each pair of the dataset to describe the location information of the data used in the study in more detail. However, since the rate of change of seabed topography, especially in the oceans far away from land, is extremely slow, requiring at least thousands or tens of thousands of years as a unit, it is of little significance to describe the acquisition time of seabed topography data in oceanographic research. Thus, the seabed topographic data obtained at different times can be regarded as on the same standard. As for the reason for using these three pairs of data in this paper, it is because we want to test our model by experimenting with the conditions of the water environment in different regions as possible. Therefore, we selected data from the Northwest Pacific, Southeast Pacific, and Southern Oceans while balancing the difficulty of data acquisition and processing with the evenness of geographic distribution. We understand that there is room for improvement.

3. In this study, we performed interpolation preprocessing on the satellite altimetry data due to the non-uniform resolution between the data pairs. We performed general bilinear interpolation on the data in ArcGIS Engine 10.6. The reason for choosing this method is to balance the amount of data computation and the accuracy of interpolation.

4. Regarding the input and output parameters of the model, what you pointed out is the case of the overall model, with the to-be-corrected satellite altimetry bathymetry data being input, and the multibeam sonar bathymetry data as the true value being output. What we have mentioned in this paper is a more intuitive explanation that focuses on the key intermediate layer in order to make the concept of loss (distance) clearer. That is, from the central perspective of the intermediate layer, the satellite altimetry and multibeam sonar bathymetry data are all input, and the corrected satellite altimetry data is output after processing and calculation in the intermediate layer. We apologize for the misunderstanding caused by the unclear expression, and have added explanations in **Line 188**.

5. In our experiments, considering the size of the data, we only divided the data set into training and testing sets (or validation/development sets in this case) in the ratio of 1:1. In future studies, if larger data sizes are applied, the validation and test sets need to be treated separately.

6. In **Line 130**, we cited *Charette et al., 2010* because we initially wanted to cite the result of global ocean mean water depth in their study. After reconsideration, we have decided to delete the reference here.

7. In **Section 3.1**, we modified the references to multibeam sonar and satellite altimetry bathymetry data. We are sorry for the mistakes here.

8. In **Line 278**, [26][29] were misplanted. We have corrected it to a GMD-compliant reference format.

9. In **Line 333**, "previous studies" has been attached with corresponding references.

10. In **Line 29** and **Line 278**, we have modified the text to NRMSE which has been improved. RMSE cannot be compared directly and needs to be converted into a normalized indicator.

11. In **Line 182**, the upper mark 40 was misplanted. We have removed it from the text.

12. We have resized **Fig.4**, **Fig.5** and **Fig.6** to meet GMD standards.

13. Professional language polish has been applied throughout the manuscript now.

Once again, we greatly appreciate your valuable comments, which have greatly benefited our manuscript. Above are our modification and responses accordingly. We would like to apologize for the mistakes and misunderstandings caused by our

carelessness, and share our views with you with a sincere heart. We are looking forward to hearing from you again.

Yours respectfully,
*Xiaowen Luo*
Sept. 1, 2022

---

## Author Comment (AC3)

**Dear Referee:**

We are grateful to receive your comments. We have carefully revised and reflected on the manuscript accordingly, and have also proposed some responses based on our opinion, which we humbly look forward to sharing with you.

- 1. Regarding the reason for choosing VGGNet over other models in this study, it is because our research is inspired by the neural style transfer algorithm for photo artistry in popular culture. It was proposed by *Gatys et al. (2015)* and has a relatively fixed paradigm, as shown in the information we added in Line 109.
- 2. Regarding the loss function, we have referred to several existing examples of neural style transfer models on *Google AI Hub*, which use the Euclidean distance function and are proven to be effective. In this study, considering that the logical relationship between pairs of bathymetry data is relatively homogeneous, we consider the Euclidean distance function to be sufficient.
- 3. For the issue of the innovativeness of the VGGNet model used in this study, we have modified it to address the characteristics and data structure features of bathymetry data. We understand that, from a machine learning perspective, our improvements to the VGGNet model may be minor. However, oceanography and geophysics offer a different perspective. As summarized in the manuscript, in current practice, the high accuracy advantage of multibeam sonar bathymetry is limited by its low spatial coverage, while the large spatial coverage advantage of satellite altimetry is limited by its low accuracy. Therefore, we believe that the practice in this study is meaningful for fusing the data of both and extracting the advantages of each. Moreover, there are only a few studies on fusion of bathymetry data using deep learning methods, so our approach is still one of the few in the field, not to mention achieving better results than previous studies. To solve this paradox, we innovatively introduce the VGGNet algorithm inspired by neural style transfer into this traditional oceanographic research topic, which can not only provide the possibility to solve the problem of high-precision mapping of global seabed topography in an efficient and sustainable way, but also provide insight into the cross-border cooperation between these two fields. Therefore, we believe that this study is of academic significance.
- 4. Regarding the bathymetry data used in this study, their grid spatial resolutions are listed in **Table 1**. Since the rate of change of seafloor topography is extremely slow, it is usually not necessary to distinguish the acquisition time of bathymetry data obtained by modern technical means in oceanographic studies.

- 5. We used bilinear interpolation in ArcGIS Engine 10.6 to standardize the resolution of different bathymetry data. Due to the simplicity of the process, we did not elaborate on it in the manuscript.
- 6. For the size of the experimental dataset, we selected the three datasets used in the experiment based on the principle of three relatively uniformly distributed seas in the Pacific Ocean, and also taking into account the ease of data acquisition and processing (after all, the choice of seas with shipborne measurements is very limited, as mentioned in the manuscript). Compared to similar previous studies, the size of the dataset used in this study is not small (>250,000 km2) and is randomly selected to accommodate simulations of many different seafloor topographic conditions.
- 7. We apologize for the errors in [26] and [29] and have updated the correct reference format. The "previous studies" mentioned in the evaluation of experimental results section of the manuscript are mostly based on the results from *Jena et al. (2012)* and *Jha et al. (2013)*. The research methods and algorithms of similar studies are described in more detail in Line 119 of the manuscript.
- 8. In Line 138, we provide some more detailed descriptions for modifying and improving the VGGNet model, including information related to the intermediate layer and the optimizer.
- 9. The figures and tables display have been reformatted to meet the need for more visualization.
- 10. Moreover, the manuscript has been professionally polished.
- 11. For minor formatting issues, the Topical Editor's opinion is to make final changes after the review phase is over.

Once again, we greatly appreciate your valuable comments, which have greatly benefited our manuscript. Above are our modification and responses accordingly. We would like to apologize for the mistakes and misunderstandings caused by our carelessness, and share our views with you with a sincere heart. We look forward to hearing from you again.

Yours respectfully, *Xiaowen Luo* Sept. 13, 2022